# Structures of the mycobacterial membrane protein MmpL3 reveal its mechanism of lipid transport

**Chih-Chia Su**[1]*, **Philip A. Klenotic**[1], **Meng Cui**[2], **Meinan Lyu**[1], **Christopher E. Morgan**[1], **Edward W. Yu**[1]*

**1** Department of Pharmacology, Case Western Reserve University School of Medicine, Cleveland, Ohio, United States of America, **2** Department of Pharmaceutical Sciences, Northeastern University School of Pharmacy, Boston, Massachusetts, United States of America

* cxs670@case.edu (C-CS); edward.w.yu@case.edu (EWY)

## Abstract

The mycobacterial membrane protein large 3 (MmpL3) transporter is essential and required for shuttling the lipid trehalose monomycolate (TMM), a precursor of mycolic acid (MA)-containing trehalose dimycolate (TDM) and mycolyl arabinogalactan peptidoglycan (mAGP), in *Mycobacterium* species, including *Mycobacterium tuberculosis* and *Mycobacterium smegmatis*. However, the mechanism that MmpL3 uses to facilitate the transport of fatty acids and lipidic elements to the mycobacterial cell wall remains elusive. Here, we report 7 structures of the *M. smegmatis* MmpL3 transporter in its unbound state and in complex with trehalose 6-decanoate (T6D) or TMM using single-particle cryo-electron microscopy (cryo-EM) and X-ray crystallography. Combined with calculated results from molecular dynamics (MD) and target MD simulations, we reveal a lipid transport mechanism that involves a coupled movement of the periplasmic domain and transmembrane helices of the MmpL3 transporter that facilitates the shuttling of lipids to the mycobacterial cell wall.

## Introduction

Tuberculosis (TB) is an airborne disease caused by the bacterium *Mycobacterium tuberculosis*. It is the leading cause of death from a single infectious agent, exceeding both malaria and HIV [1,2]. The emergence and spread of multidrug-resistant TB (MDR-TB) presents an increasingly difficult therapeutic challenge, thus a serious threat to our global public health. The lethality of TB combined with its multidrug-resistant capacity has already transformed this disease into a worldwide problem.

The unique architecture of the mycobacterial cell wall plays a predominant role in *M. tuberculosis* pathogenesis. This complex cell wall layer provides a strong barrier to protect *M. tuberculosis* against the host immune response as well as clinically relevant antibiotics. The outer membrane of *M. tuberculosis* is distinguished by the hallmark lipid mycolic acid (MA), a specific 2-alkyl-3-hydroxyl long-chain fatty acid. MA is shuttled across the inner membrane as trehalose monomycolate (TMM), which is then either incorporated into trehalose dimycolate

(PDB) and EMD-22724 (EMBD) for MmpL3-ND; 7K8B (PDB) and EMD-22725 (EMDB) for MmpL3-GDN; 7K8C (PDB) and EMD-22726 (EMDB) for MmpL3-TMM I; 7K8D (PDB) and EMD-22728 (EMDB) for MmpL3-TMM II; 7N6B (PDB) and EMD-24206 (EMDB) for MmpL3-TMM III; and 7K7M (PDB) for MmpL3773-T6D I and MmpL3773-T6D II.

**Funding:** This work was funded by the National Institues of Health R01AI145069 (to EWY) (https://taggs.hhs.gov/Detail/AwardDetail?arg_AwardNum=R01AI145069&arg_ProgOfficeCode=104), in part, supported by the National Cancer Institute's National Cryo-EM Facility at the Frederick National Laboratory for Cancer Research under contract HSSN261200800001E. A portion of this research was supported by NIH grant U24GM129547 and performed at the PNCC at OHSU and accessed through EMSL (grid.436923.9), a DOE Office of Science User Facility sponsored by the Office of Biological and Environmental Research. The funders had no role in study design, data collection and analysis, decision to publish, or preparation of the manuscript.

**Competing interests:** The authors have declared that no competing interests exist.

**Abbreviations:** cryo-EM, cryo-electron microscopy; CTF, contrast transfer function; DDM, n-dodecyl-β-D-maltoside; EDTA, ethylenediaminetetraacetic acid; GB, Generalized Born; GDN, glycol-diosgenin; MA, mycolic acid; mAGP, mycolyl arabinogalactan peptidoglycan; MD, molecular dynamics; MDR-TB, multidrug-resistant TB; MM-GBSA, molecular mechanics generalized Born surface area; MmpL, mycobacterial membrane protein large; MmpL3-ND, MmpL3-nanodisc; MR, molecular replacement; OGNG, octyl glucose neopentyl glycol; PCA, principal component analysis; PE, phosphatidylethanolamine; PME, particle mesh Ewald; PMSF, phenylmethanesulfonyl fluoride; RMS, root mean square; RMSD, root mean square deviation; TB, tuberculosis; RMSF, root mean square fluctuation; TDM, trehalose dimycolate; TMD, target MD; TMM, trehalose monomycolate; T6D, trehalose 6-decanoate.

(TDM) or covalently linked to the arabinogalactan-peptidoglycan layer as mycolyl arabinoga-lactan peptidoglycan (mAGP). Recently, it has been demonstrated that mycobacterial membrane protein large (MmpL) transporters are critical for mycobacterial physiology and pathogenesis by shuttling fatty acids and lipid components to the mycobacterial cell wall. Of the 13 *M. tuberculosis* MmpLs, only MmpL3 is shown to be capable of exporting TMMs [3,4]. A detailed analysis suggested that MmpL3 is a TMM flippase [5] that is absolutely essential for transporting TMMs across the cell membrane. Because of the requirement of MmpL3 for mycobacterial cell wall biogenesis, this membrane protein has been a target of several potent anti-TB agents [6–13].

Interestingly, it has been observed that *M. tuberculosis mmpL3* is able to rescue the utility of the *Mycobacterium smegmatis mmpl3* null mutant [3]. Both viability and lipid transport studies have shown that these 2 *mmpL3* orthologs are functionally interchangeable. We recently determined X-ray structures of the *M. smegmatis* MmpL3 transporter, revealing a monomeric molecule that is structurally distinct from all known bacterial membrane proteins [14]. This structural information indeed agrees with the structure of MmpL3 in the presence of a variety of inhibitors [15]. The MmpL3 transporter contains 12 transmembrane helices and 2 periplasmic flexible loops, posing a plausible pathway for substrate transport. Using native mass spectrometry, we found that MmpL3 is a promiscuous membrane protein, capable of binding a variety of lipids, including TMM, phosphatidylethanolamine (PE), phosphatidylglycerol, and cardiolipin, all with dissociation constants within the micromolar range [14]. Our previous experimental data highlight a possible role for MmpL3 in shuttling different lipids across the membrane. To further elucidate the molecular mechanism of lipid translocation across the mycobacterial membrane via the MmpL3 transporter, we here present 7 structures of *M. smegmatis* MmpL3, either alone or bound with lipid moieties, using single-particle cryo-electron microscopy (cryo-EM) and X-ray crystallography. We also investigate the functional dynamics of this transporter using molecular dynamics (MD) simulation. Together, combined with new structural information, target MD simulation provides us with a pathway for lipid transport via the MmpL3 membrane protein.

## Results

### Cryo-EM structure of MmpL3 reconstituted in nanodiscs

We expressed the full-length *M. smegmatis* MmpL3 transporter by cloning *mmpL3* into the *Escherichia coli* expression vector pET15b, with a 6xHis tag at the carboxyl terminus to generate pET15bΩ*mmpL3*. The MmpL3 membrane protein was overproduced in *E. coli* BL21(DE3) *ΔacrB* cells. We delipidated and purified this membrane protein using a $Ni^{2+}$-affinity column. We then reconstituted the delipidated, purified MmpL3 transporter into lipidic nanodiscs. This MmpL3-nanodisc (MmpL3-ND) sample was further purified using a Superose 6 column to separate the MmpL3-ND complex from empty nanodiscs. The structure of MmpL3-ND was solved using single-particle cryo-EM. The reconstituted sample led to a cryo-EM map at a nominal resolution of 3.65 Å (Fig 1A, S1 Table and S1 Fig). Density modification [16] allowed us to improve the resolution of the map to 3.00 Å.

The cryo-EM structure of MmpL3-ND indicates a monomeric molecule that spans the inner membrane and protrudes into the periplasm (Fig 1B). Like the X-ray structure of the detergent-solubilized MmpL3 protein, the carboxyl terminal end of MmpL3 is absent in the cryo-EM structure. MmpL3 consists of 12 transmembrane helices (TMs 1 to 12) and 2 periplasmic subdomains (PDs 1 and 2). Overall, the conformation of the cryo-EM structure of MmpL3-ND is comparable to that of the X-ray structure of MmpL3773-PE, although there are some conformational differences between these 2 structures. Superimposition of 710 Cα

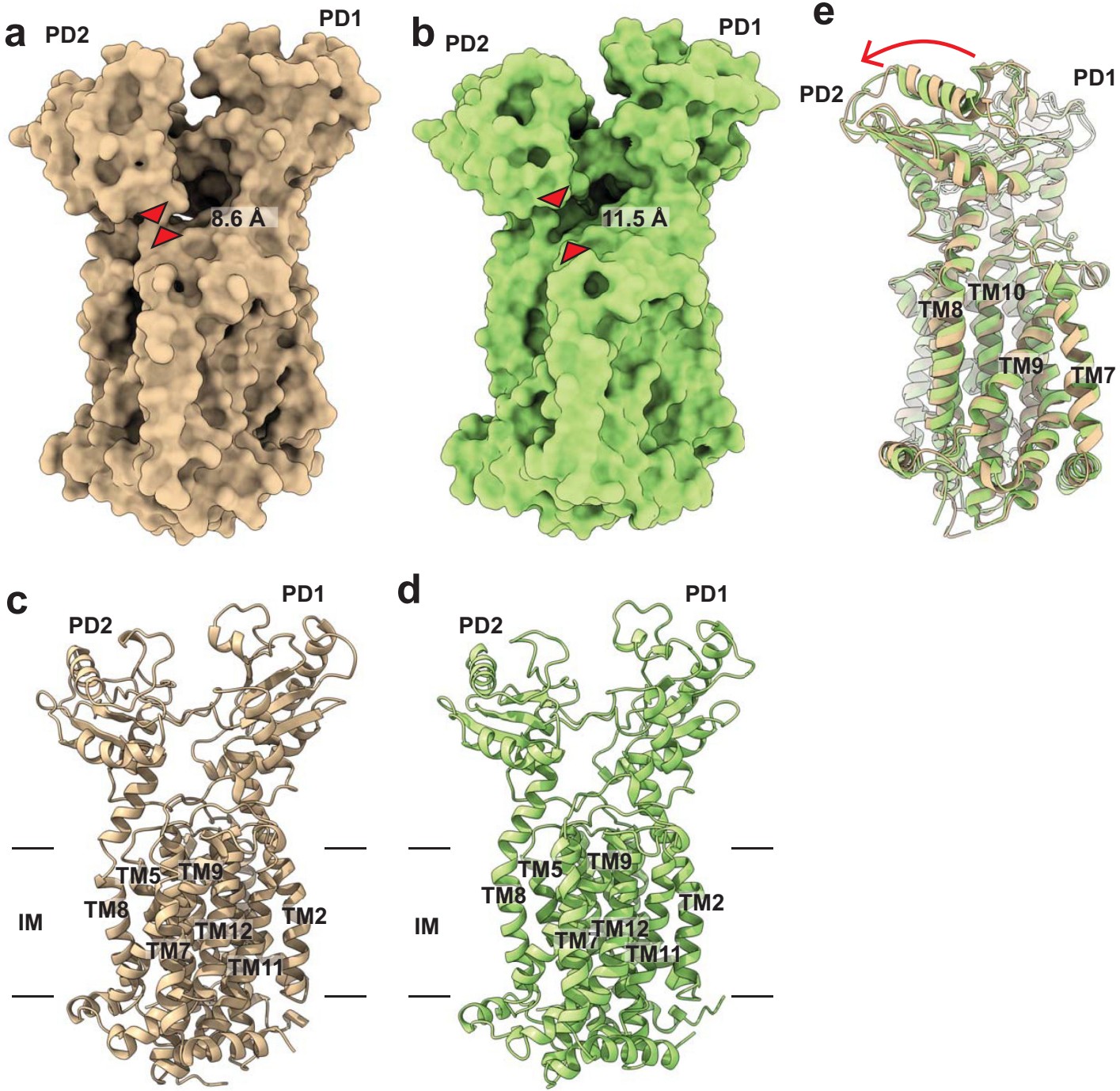

**Fig 1. Structures of the *M. smegmatis* MmpL3 transporter.** (a) Side view of the surface representation of MmpL3-ND. The narrowest region of the channel created by the MmpL3 membrane protein, measured between the Cα atoms of residues S423 and N524 (red triangles), is 8.6 Å. (b) Ribbon diagram of MmpL3-ND viewed in the membrane plane. (c) Side view of the surface representation of MmpL3-GDN. The narrowest region of the channel created by the MmpL3 membrane protein, measured between the Cα atoms of residues S423 and N524 (red triangles), is 11.5 Å. (d) Ribbon diagram of MmpL3-GDN viewed in the membrane plane. (e) Superimposition of the structures of MmpL3-ND and MmpL3-GDN. This superimposition suggests that the major difference between these 2 structures is the location of PD2, which rotates by approximately 6° in a rigid body fashion (red arrow) when compared the MmpL3-GDN structure to that of MmpL3-ND. MmpL3, mycobacterial membrane protein large 3; MmpL3-GDN, MmpL3-glycol-diosgenin; MmpL3-ND, MmpL3-nanodisc.

atoms of these 2 structural models gives rise to a root mean square deviation (RMSD) of 1.5 Å (S2 Fig). The conformation of MmpL3-ND is distinct in that the gap between subdomains PD1 and PD2 is larger when compared with that of MmpL3$_{773}$-PE. The volume of the periplasmic central cavity formed between PD1 and PD2 was measured to be 1,794 Å$^3$ in MmpL3-ND. The corresponding volume of this cavity in the MmpL3$_{773}$-PE structure is 946 Å$^3$, only half of the volume observed in the cryo-EM structure. These volumes were computed using the program CASTp 3.0 [17]. Based on the structural information, the increased volume of the periplasmic central cavity of MmpL3-ND is in part caused by the increased separation of PD1 and PD2. However, the expansion of this central volume is also related to the change in conformation of several transmembrane helices, which accommodates for the rearrangement of the periplasmic domain. It is observed that TM7, TM9, TM10, TM11, and TM12 slightly shift toward the cytoplasm by approximately 2 to 3 Å (S2 Fig). This shift, in turn, helps expand the capacity of the periplasmic central cavity.

## Cryo-EM structure of MmpL3 embedded in detergent micelles

Overall, the cryo-EM structure of MmpL3 embedded in nanodiscs is in good agreement with that of the X-ray structure of MmpL3$_{773}$-PE, although these 2 structures represent different transient states of the membrane protein. It is uncertain if the cryo-EM structure of this membrane protein would be different in a detergent environment. To answer this question, we delipidated and purified MmpL3 in a solution containing 0.01% glycol-diosgenin (GDN) detergent. We then determined the cryo-EM structure of MmpL3 surrounded by GDN detergent micelles at a nominal resolution of 2.94 Å (MmpL3-GDN) (Fig 1C, S1 Table and S3 Fig). The density modification program [16] allowed us to improve the resolution of the cryo-EM map to 2.40 Å (Fig 1D). The structures of MmpL3-GDN and MmpL3-ND largely resemble each other. However, there are conformational differences between the two. Superimposition of these 2 structures gives rise to an RMSD of 1.4 Å (for 710 Cα atoms), suggesting that these conformations are distinct from each other (Fig 1E). The calculated volume of the periplasmic central cavity of GDN solubilized MmpL3 using CASTp 3.0 [17] is 1,716 Å$^3$, which is similar, albeit smaller, to that of the protein embedded in nanodiscs. A detailed inspection on these 2 structures suggests that there is a relative rotational motion involved between the periplasmic subdomains PD1 and PD2 of the MmpL3 membrane protein, where PD2 is found to rotate by 6˚ in relation to PD1 of MmpL3-GDN when compared with the structure of MmpL3-ND (Fig 1E).

## Cryo-EM structures of MmpL3 in the presence of TMM

Since cryo-EM structures of the transporter surrounded by nanodiscs or detergent micelles are in good agreement with each other, we decided to determine a cryo-EM structure of MmpL3 embedded in nanodiscs in the presence of TMM lipids. Surprisingly, we found the existence of 2 distinct conformations of single-particle images in the sample. These images led us to resolve 2 different cryo-EM structures of MmpL3 at nominal resolutions of 4.26 Å and 4.33 Å (MmpL3-TMM I and MmpL3-TMM II), respectively (Fig 2A and 2B, S1 Table and S4 Fig). Additional density modification [16] permitted us to improve the resolution of these 2 structures to 3.30 Å (Fig 2C and 2D). The conformations of the 2 structures are markedly different from each other (Fig 2E). They are also distinct from those of MmpL3-ND and MmpL3-G6D. Pairwise superimpositions of the structures of MmpL3-TMM I and MmpL3-TMM II to that of MmpL3-ND result in RMSD values of 1.7 Å and 2.3 Å (for 710 Cα atoms each), respectively. Unfortunately, we did not observe any bound TMM molecules within these 2 MmpL3 structures. In MmpL3-TMM I, the volume of the periplasmic cavity was measured to be 1,303 Å$^3$,

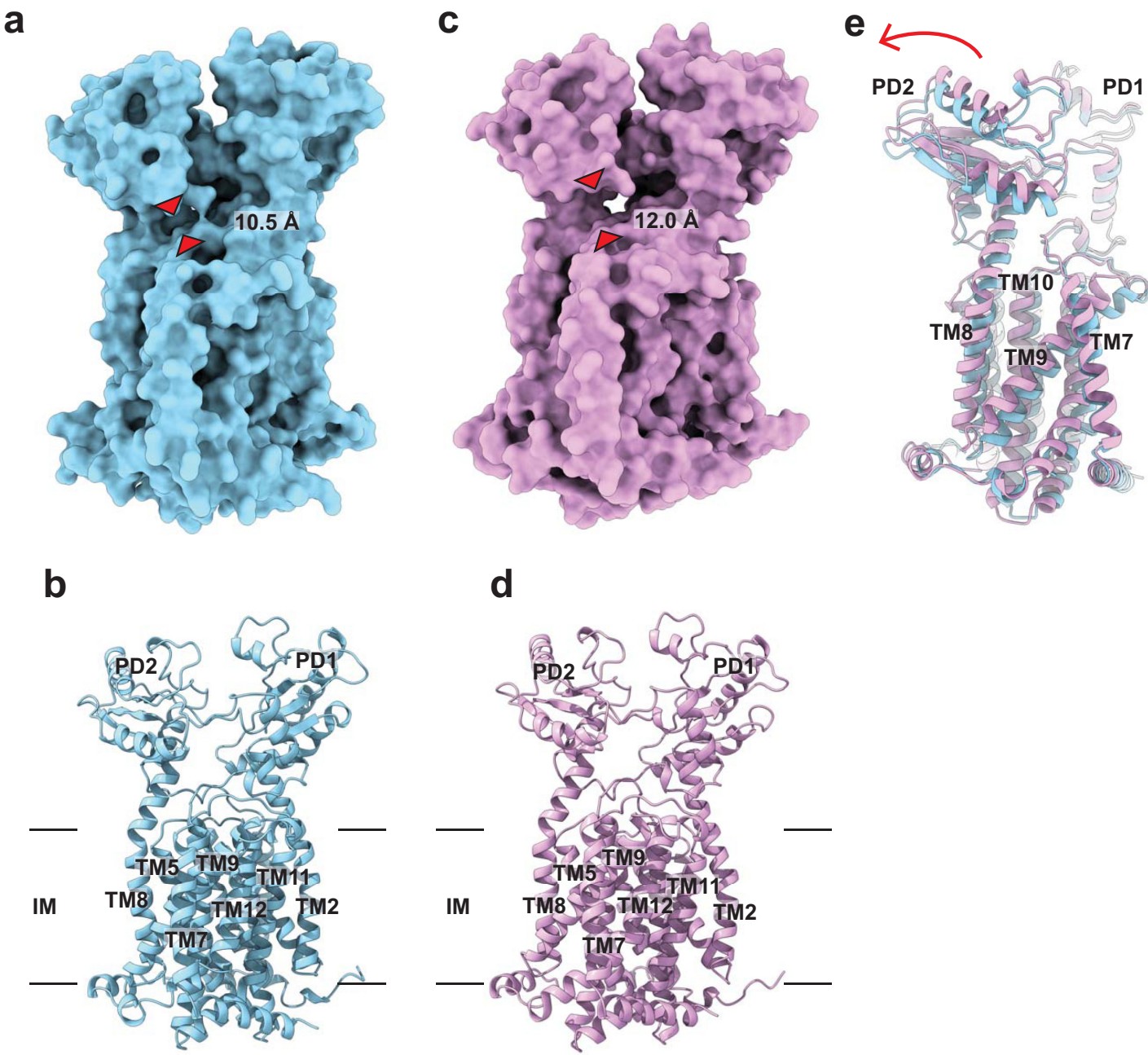

**Fig 2. Structures of MmpL3-TMM I and MmpL3-TMM II.** (a) Side view of the surface representation of MmpL3-TMM I. The narrowest region of the channel created by the MmpL3 membrane protein, measured between the Cα atoms of residues S423 and N524 (red triangles), is 10.5 Å. (b) Side view of the surface representation of MmpL3-GDN. The narrowest region of the channel created by the MmpL3 membrane protein, measured between the Cα atoms of residues S423 and N524 (red triangles), is 12.0 Å. (c) Ribbon diagram of MmpL3-TMM I viewed in the membrane plane. (d) Ribbon diagram of MmpL3-TMM II viewed in the membrane plane. (e) Superimposition of the structures of MmpL3-TMM I and MmpL3-TMM II. This superimposition suggests that there is a drastic change in conformation of the transmembrane helices, including TMs 7 and 10, in addition to the rigid body movement of subdomain PD2. MmpL3, mycobacterial membrane protein large 3; MmpL3-GDN, MmpL3-glycol-diosgenin; MmpL3-TMM, MmpL3-trehalose monomycolate; TM, transmembrane.

which is 491 Å$^3$ smaller than that of apo MmpL3-ND and 357 Å$^3$ larger than that of MmpL3$_{773}$-PE. However, the calculation of the volume of this periplasmic central cavity in MmpL3-TMM II gave rise to 2,212 Å$^3$, which is approximately 420 Å$^3$ and 1,266 Å$^3$ larger than those of MmpL3-ND and MmpL3$_{773}$-PE. These volumes were computed using CASTp 3.0 [17]. A comparison of the 3 nanodisc-embedded structures of MmpL3 suggests that there is a drastic conformational change of the transmembrane helices in addition to the rigid body movement of subdomain PD2. This change can be interpreted as a shift in the relative distance between TM2 and TM8. This change also alters the relative positions of several TM helices, including TMs 7 to 12.

## X-ray structures of the MmpL3$_{773}$-T6D complexes

In addition to elucidating the structures of MmpL3 via cryo-EM, we crystallized the delipidated, purified MmpL3$_{773}$ transporter, which contains residues 1 to 773, in the presence of trehalose 6-decanoate (T6D). T6D is an ideal ligand to mimic the TMM lipid, as it contains a trehalose moiety and is structurally similar to TMM. Crystals of MmpL3$_{773}$ bound with T6D diffracted X-rays to a resolution of 3.34 Å (Fig 3A–3D, S2 Table and S5 Fig). Surprisingly, the data indicate that each asymmetric unit of the crystal contains 2 independent MmpL3$_{773}$ monomers, where the conformations (MmpL3$_{773}$-T6D I and MmpL3$_{773}$-T6D II) are different from each other (Fig 3E). These 2 structures are also similar but distinct from that of MmpL3$_{773}$-PE. Pairwise superimpositions of the structure of MmpL3$_{773}$-PE to those of MmpL3$_{773}$-T6D I and MmpL3$_{773}$-T6D II give rise to RMSD values of 1.4 Å and 1.5 Å (for 710 Cα atoms each), respectively. Interestingly, the structures of MmpL3$_{773}$-T6D I and MmpL3$_{773}$-T6D II are quite different both in the periplasmic and transmembrane domains. Particularly, it is observed that TM9 of MmpL3$_{773}$-T6D II tilts away from the innermost surface of the cytoplasmic membrane when compared with that of MmpL3$_{773}$-T6D I. Consistent with the structure of MmpL3$_{773}$-PE, each MmpL3$_{773}$-T6D structure is found to create 2 substrate binding sites that are situated at cavities formed by TM7-TM10 in the transmembrane region and between PD1 and PD2 of the periplasmic domain, respectively. Each binding site is occupied by a T6D molecule.

A large density corresponding to the first bound T6D molecule (T6D$_1$) was observed in the pocket surrounded by TMs 7 to 10 in each structure. In the MmpL3-T6D I structure, residues S423, L424, Q554, F561, L564, A568, and I636 are within 4.5 Å of the bound T6D$_1$, performing hydrophobic or polar interactions with this ligand (Fig 3C). In addition, N524 and H558 are situated approximately 5.3 Å away from the trehalose headgroup, interacting with this bound T6D$_1$ molecule via polar interaction. Interestingly, at least half of these contacting residues belong to TM8, suggesting that TM8 may be an important transmembrane helix for recognizing and shuttling TMM to the periplasm. In the MmpL3-T6D II structure, residues that are found to be important for T6D$_1$ binding are S423, L424, Q554, F561, L564, and I636, in which the composition of these residues are very similar to that of MmpL3-T6D I (Fig 3D).

The second large density corresponding to the other bound T6D (T6D$_2$) was observed in the central periplasmic cavity between PD1 and PD2. For MmpL3-T6D I, this bound T6D$_2$ molecule is surrounded by residues Q40, S41, F43, R63, D166, L171, L178, and T549, which are within 4.5 Å of bound T6D$_2$, performing electrostatic or polar interactions to bind the ligand (Fig 3C). Similarly, the MmpL3 transporter utilizes residues Q40, S41, F43, Y44, D58, R63, L171, L178, N524, and T549 to contact and participate in electrostatic interactions with the trehalose headgroup of T6D$_2$ in the MmpL3-T6D II (Fig 3D). One obvious distinction between these 2 structures are the exact locations of their bound T6Ds. The 2 bound T6Ds in the MmpL3-T6D II structure seemingly shift upward and move away from the surface of the

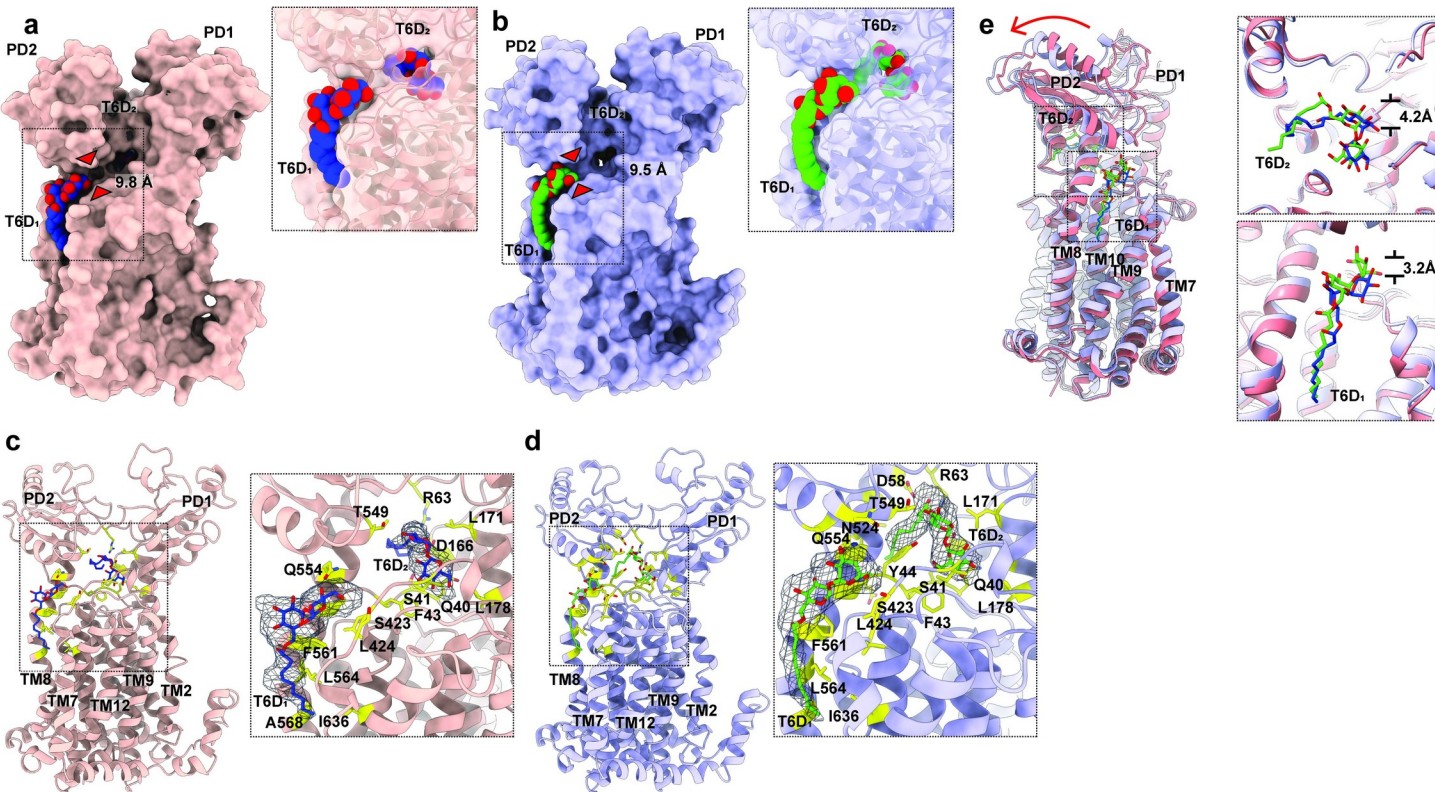

**Fig 3. Structures of MmpL3-T6D I and MmpL3-T6D II.** (a) Side view of the surface representation of MmpL3-T6D I. The narrowest region of the channel within MmpL3, as measured between the Cα atoms of residues S423 and N524 (red triangles), is 9.8 Å. The 2 bound T6D molecules (T6D$_1$ and T6D$_2$; insert) are in blue and red spheres, respectively. (b) Side view of the surface representation of MmpL3-T6D II. The narrowest region of the channel within MmpL3, as measured between the Cα atoms of residues S423 and N524 (red triangles), is 9.5 Å. The 2 bound T6D molecules (T6D$_1$ and T6D$_2$; insert) are in green and red spheres. (c) Ribbon diagram of MmpL3-T6D I viewed in the membrane plane. The F$_o$—F$_c$ electron density maps of the 2 bound T6D molecules are contoured at 3σ (insert). The bound T6D$_1$ and T6D$_2$ molecules are shown as sticks (blue, carbon; red, oxygen). (d) Ribbon diagram of MmpL3-T6D II viewed in the membrane plane. The F$_o$—F$_c$ electron density maps of the 2 bound T6D molecules are contoured at 3σ (insert). The bound T6D$_1$ and T6D$_2$ molecules are shown as sticks (green, carbon; red, oxygen). (e) Superimposition of the structures of MmpL3-T6D I and MmpL3-T6D II. This superimposition suggests that there is a drastic change in conformation of subdomain PD2 when compared between these 2 structures. In addition, the locations of the 2 bound T6Ds are quite distinct. MmpL3, mycobacterial membrane protein large 3; MmpL3-T6D, MmpL3-trehalose 6-decanoate; T6D, trehalose 6-decanoate.

membrane by 3.2 Å (at the transmembrane binding site) and 4.2 Å (at the periplasmic binding site) in comparison with the corresponding bound T6Ds in the MmpL3-T6D I structure (Fig 3E). It appears that these 2 structures capture different transient states of substrate transport mediated by MmpL3. Interestingly, these 2 structures suggest that the interactions between MmpL3 and T6D seem to shift from a more hydrophobic nature in the transmembrane region to a more electrostatic nature at the periplasmic domain. It is likely that the phosphate head-groups of the outer leaflet surface of the inner membrane provide electrostatic interactions to the bound T6D molecule in the substrate binding site of the transmembrane region.

## Cryo-EM structure of the MmpL3-TMM complex

After obtaining crystal structures of the MmpL3$_{773}$-T6D complexes, we rationalized that TMMs should be bound at locations similar to the T6D binding sites. We then repeated the cryo-EM experiment of MmpL3 with a 2-fold increase in the concentration of TMM in the nanodiscs, with the expectation that we could identify the TMM binding sites within the transporter. We reconstituted the delipidated, purified MmpL3 membrane protein into these

nanodiscs and collected single-particle cryo-EM images. The three-dimensional reconstitution of MmpL3 led to a 2.66 Å-resolution cryo-EM map. Density modification [16] enabled us to further improve the resolution, resulting in resolving the MmpL3-TMM complex structure (MmpL3-TMM III) to a resolution of 2.20 Å (Fig 4A–4D, S1 Table and S6 Fig). Pairwise super-impositions of 710 Cα atoms of the structure of MmpL3-TMM III to those of the above 6 MmpL3 structures give rise to RMSD values ranging from 0.6 Å to 2.4 Å. These superimposi-tions also indicate that the conformation of MmpL3-TMM III is nearly identical to that of MmpL3$_{773}$-T6D II (RMSD = 0.6 Å), suggesting that these 2 MmpL3 structures represent a very similar transient state of the transporter.

Based on the MmpL3-TMM III structure, we identified 2 extra densities, which indicate that there are 2 TMM molecules bound by the MmpL3 transporter (Fig 4A and 4B). As postu-lated, the locations of these 2 bound TMM molecules coincide with those of the T6D binding sites. The first bound TMM molecule (TMM$_1$) was observed within the pocket at the outer leaflet of the cytoplasmic membrane. This pocket is created by TMs 7 to 10 of the transmem-brane region. The second bound TMM lipid (TMM$_2$) was found to sandwich between PD1 and PD2 of the periplasmic domain.

At the outer leaflet of the cytoplasmic membrane, TMM$_1$ is bound in such a way that the trehalose headgroup is located at the surface of the membrane, presumably interacting with the phosphate headgroups of phospholipids, leaving the elongated hydrophobic tail of carbon chains contacting the hydrocarbon chains of phospholipids within the membrane. The length of the elongated tail of TMM is quite substantial such that it almost spans the entire cyto-plasmic membrane of the bacterium. Within 4.5 Å of bound TMM$_1$, the MmpL3 residues I416, L419, L422, S423, L424, I557, F561, L564, A568, I572, V573, T576, I590, A593, L594, A597, L598, L600, M604, I632, I636, and W640 are responsible for the binding (Fig 4B). In addition, the backbone oxygen of L422 forms a hydrogen bond with TMM$_1$ to further stabilize the binding. It should be noted that most of these amino acids are incorporated into TM8 and TM9, suggesting the importance of these 2 TMs for recognizing TMM.

As mentioned, the bound TMM$_2$ molecule was found at the central cavity formed by sub-domains PD1 and PD2 of the periplasmic domain of MmpL3. The 2 extended loops, which run across PD1 and PD2, are engaged in housing this lipid molecule. Surprisingly, the orienta-tion of this bound TMM$_2$ molecule is more or less antiparallel to that of bound TMM$_1$ found in the transmembrane region. Therefore, the hydrophobic carbon tail of bound TMM$_2$ at the periplasmic domain was observed to point toward the cell wall skeleton. The binding of this TMM$_2$ lipid is extensive. Within 4.5 Å of bound TMM$_2$, there are at least 31 residues engaged in binding this lipid. These residues are Q40, Y44, D64, T66, S67, V70, V109, K113, A114, V122, M125, F134, S136, L171, L174, A175, L178, S300, I427, E429, Q442, F445, F452, R453, T454, E455, T488, P490, K499, Q517, and T549, which provide electrostatic and hydrophobic interactions to bind this TMM$_2$ molecule (Fig 4B). Particularly, E429 and R453 also form hydrogen bonds with TMM$_2$ to strengthen the binding.

## Computational simulations of the MmpL3 transporter

One of the major differences between these 7 MmpL3 cryo-EM and X-ray structures is the conformation of the periplasmic subdomain PD2. It appears that PD2 is able to perform a rigid body rotational motion with respect to subdomain PD1, transitioning from one confor-mational state to the other. In view of the 7 structures, the transmembrane helices TM7 to TM10 are also found to shift their locations, seemingly accompanying the rotational motion of PD2. It is possible that these 2 major conformational changes could couple with each other to facilitate the translocation of lipids across the transporter. Based on the structures of MmpL3,

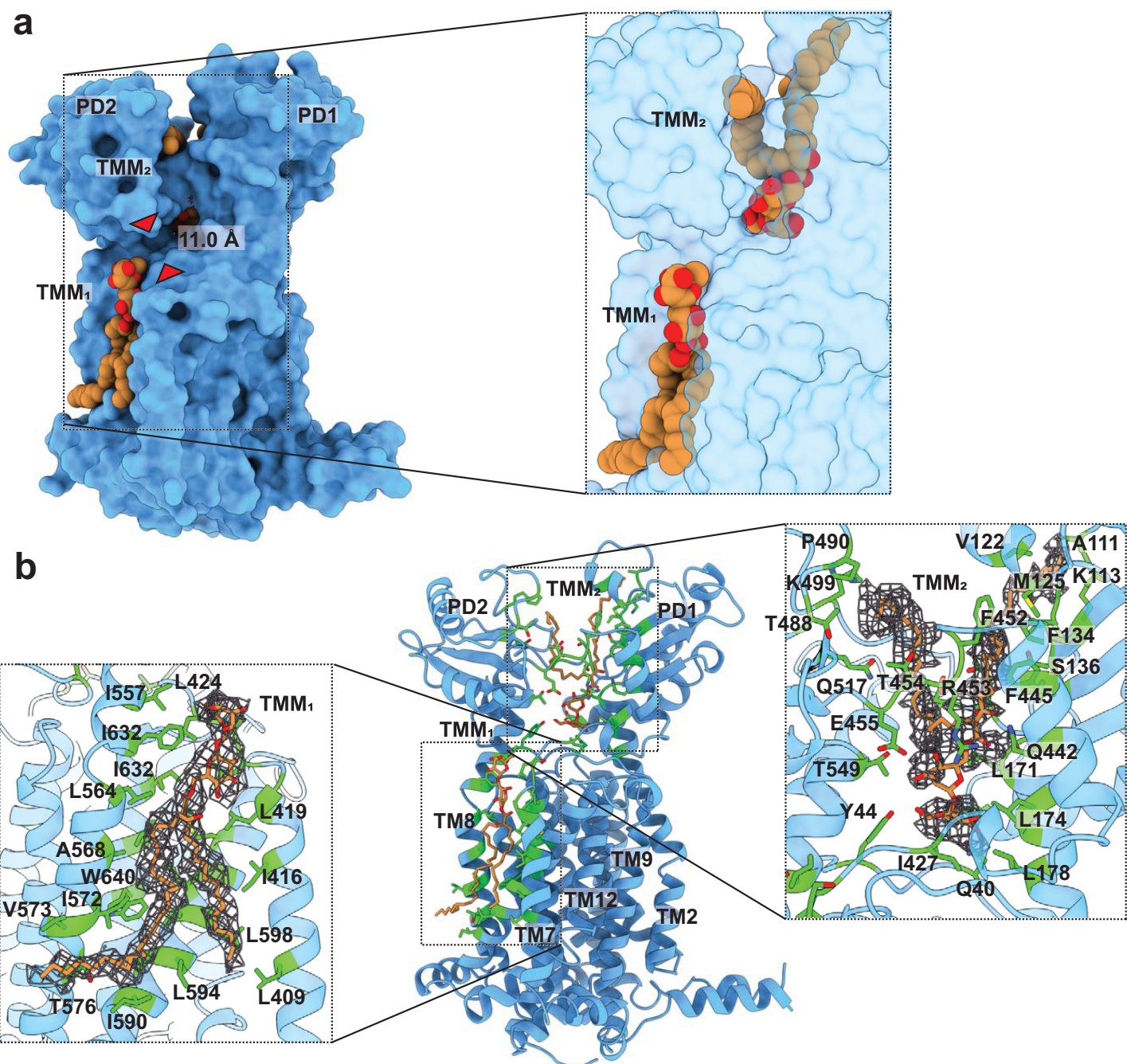

**Fig 4. Structures of the MmpL3-TMM complex.** (a) Side view of the surface representation of MmpL3-TMM III. The narrowest region of the channel created by the MmpL3 membrane protein, measured between the Cα atoms of residues S423 and N524 (red triangles), is 11.0 Å. The 2 bound TMM lipids (TMM$_1$ and TMM$_2$) are in orange and red spheres, respectively. (b) Ribbon diagram of MmpL3-TMM III viewed in the membrane plane. The 2 bound TMM molecules are in orange sticks. The binding residues I416, L419, L422, S423, L424, I557, F561, L564, A568, I572, V573, T576, I590, A593, L594, A597, L598, L600, M604, I632, I636, and W640 for TMM$_1$, as well as the binding residues Q40, Y44, D64, T66, S67, V70, V109, K113, A114, V122, M125, F134, S136, L171, L174, A175, L178, S300, I427, E429, Q442, F445, F452, R453, T454, E455, T488, P490, K499, Q517, and T549 for TMM$_2$ are colored green. MmpL3, mycobacterial membrane protein large 3; MmpL3-TMM, MmpL3-trehalose monomycolate.

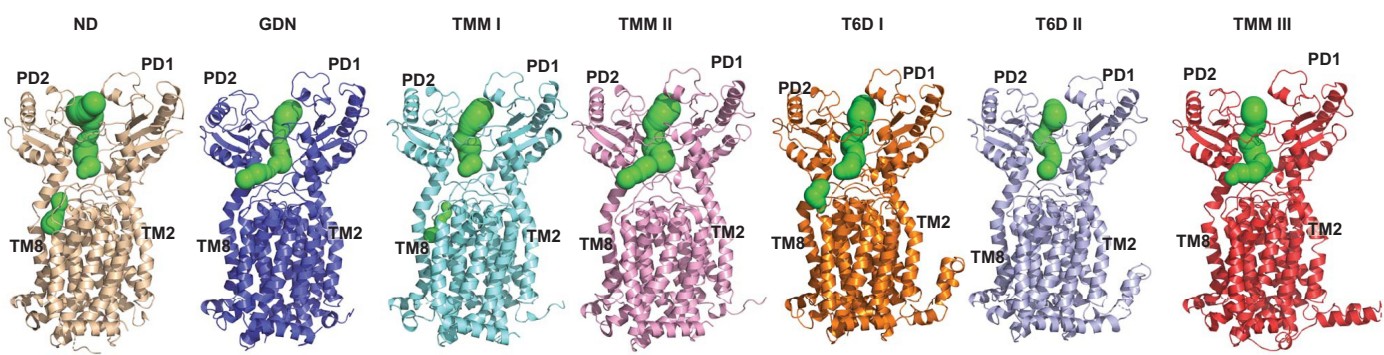

**Fig 5. Channel created by the MmpL3 transporter.** This figure indicates that the structures of MmpL3-ND, MmpL3-T6D I, MmpL3-T6D II, MmpL3-TMM I, and MmpL3-TMM III form truncated channels, where S423 and N524 are responsible to close these channels. However, the structures of MmpL3-GDN and MmpL3-TMM II represent the fully open channel conformations. MmpL3, mycobacterial membrane protein large 3; MmpL3-GDN, MmpL3-glycol-diosgenin; MmpL3-ND, MmpL3-nanodisc; MmpL3-TMM, MmpL3-trehalose monomycolate; MmpL3-T6D, MmpL3-trehalose 6-decanoate.

residues S423 and N524 form the narrowest region of the channel. The distances between the Cα atoms of these 2 residues range from 8.6 Å to 12.0 Å among the 7 structures. There is a chance that S423 and N524 may form a gate of the channel that is able to open and close in order to advance the transport cycle (Fig 5) as indicated in our MD simulations (see below).

The rigid body rotational motion of subdomain PD2 also alters the relative locations of TM2 and TM8. The distances between TM2 and TM8 at the periplasmic surface of the inner membrane, as measured between the Cα atoms of residue T181 of TM2 and residue D542 of TM8, range from 31.6 Å to 35.5 Å in these 7 structures. This change can be interpreted as a shift in the relative distance between TM2 and TM8. Apparently, this change allows for the alteration of the size of the periplasmic central cavity to accommodate the lipid ligand.

Our MmpL3 structures depict that subdomain PD2 can perform a rigid body rotational motion with respect to subdomain PD1 (Fig 6A). To elucidate lipid transport via the MmpL3 membrane protein, we performed MD simulations based on the X-ray structure of MmpL3$_{773}$-T6D I. Principal component analysis (PCA) indicates that the first eigenvector, which depicts the most important motion extracted from the MD simulation trajectory, corresponds to a rotational motion of the periplasmic subdomain PD2 in relation to PD1 (Fig 6B). This result is indeed in good agreement with our cryo-EM and X-ray structures, suggesting that the major conformational difference of the transporter structures comes from subdomain PD2, where its relative locations can be interpreted as snapshots of a rigid body rotation of this subdomain. The narrowest region of the elongated channel formed by the MmpL3 transporter is surrounded by a flexible loop constituting residues R523, N524, and D525 on one side, and another loop containing residues S423, L424, G425, and G426 on the other side of the channel. These residues may be important for controlling the opening and closing of this channel by coupling with the dynamic motion of the periplasmic subdomain PD2 (Fig 6C and 6D). This result is also in line with our experimental structural data that S423 and N524 form the narrowest region of the channel.

Based on the crystal structures of the MmpL3-T6D complex, there are 2 T6D-binding sites, one located at the cleft surrounded by TMs 7 to 10 and the other within the large central cavity between PD1 and PD2. Previously, we hypothesized that the MmpL3 transporter takes up lipids from the outer leaflet of the cytoplasmic membrane via the channel, spanning the outer leaflet of the cytoplasmic membrane from cleft between TMs 7 to 10 up to the central cavity of the periplasmic domain between PD1 and PD2 [14]. Our MD simulations agree with this and indicate that there are 4 particular residues, R523, K528, H558, and F561, lining the wall of the

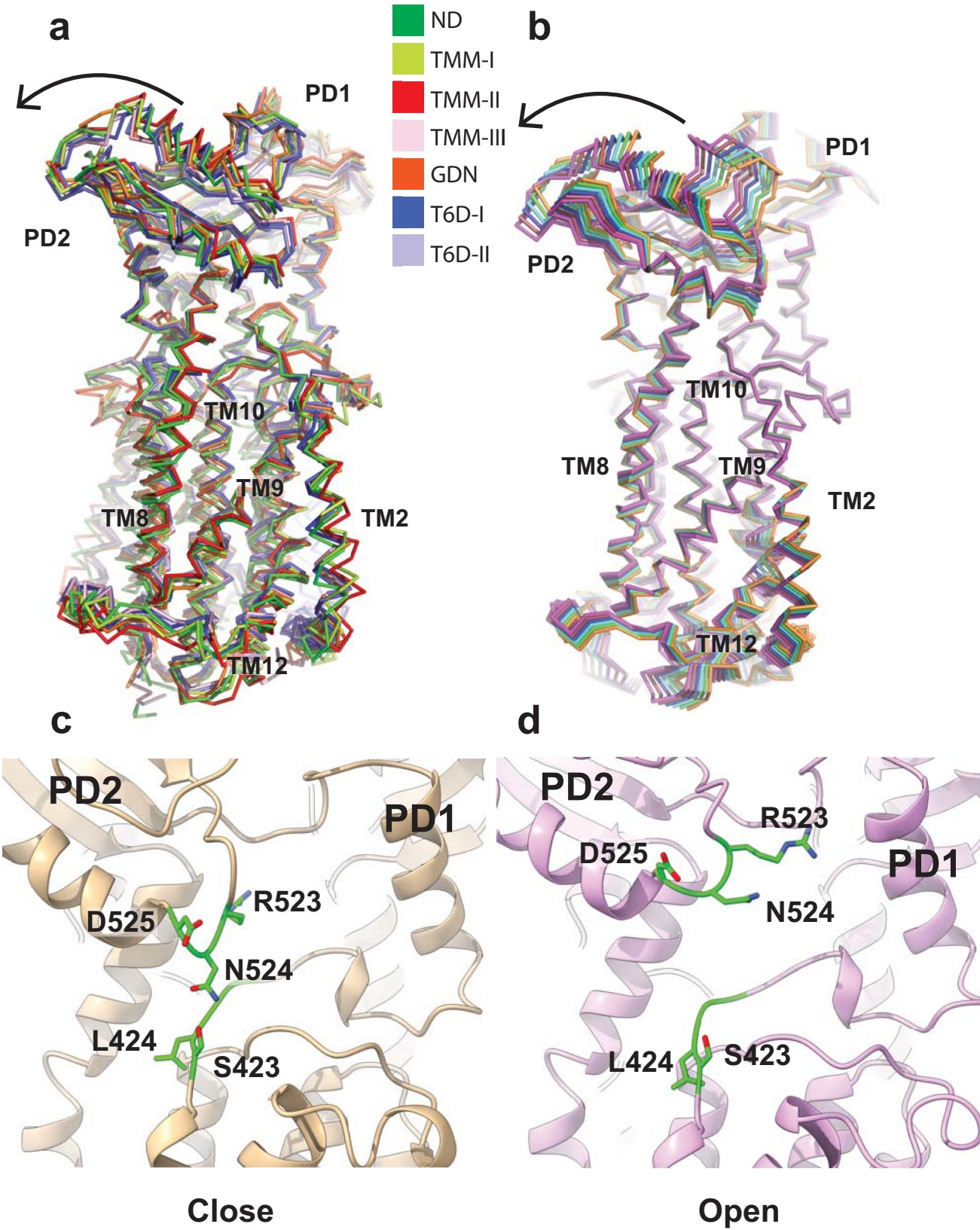

**Fig 6. Conformational flexibility of the MmpL3 transporter.** (a) Superimposition of the 7 structures of MmpL3. The structures indicate that PD2 is able to perform a rigid body rotational motion, transitioning from one conformational state to the other. The transmembrane helices, particularly TM7–TM12, are also found to shift their locations, accompanying the rotational motion of PD2. (b) MD simulations of the MmpL3 transporter. Consistent with the 7 structures of MmpL3, the first eigenvector based on PCA suggests that the major motion of MmpL3 is the rigid body rotation of the PD2 subdomain. (c and d) The 2 flexible loops 423–426 and 523–525 create the narrowest region of the channel. These residues may be important for controlling the (c) closing and (d) opening of this channel by coupling with the dynamic motion of the periplasmic subdomain PD2. Residues S423, L424, R523, N524, and D525 are indicated as green sticks. MD, molecular dynamics; MmpL3, mycobacterial membrane protein large 3; PCA, principal component analysis.

channel. These 4 residues may be important for directing the bound T6D ligand from the transmembrane to the periplasmic lipid-binding sites. Notably, H558 and F561 are found to perform critical intact interactions with bound T6D at the transmembrane region (S7 Fig). In contrast, R523 and K528 engage in dynamic interactions with the trehalose headgroup of T6D (S7 Fig). These dynamic interactions, together with the motion of subdomain PD2, could be the key to facilitate lipid transport from the membrane region to the periplasmic domain of the MmpL3 transporter.

It appears that 3 out of 4 of these residues, including K528, H558, and F561, are conserved among the mycobacterial species (S8 Fig). We then used the approach of molecular mechanics generalized Born surface area (MM-GBSA) [18] to conduct computational alanine scanning experiment on residues K528, H558, and F561 of the MmpL3 transporter to study the effects of mutation on the binding affinity of the T6D ligand. We observed that the binding affinity for T6D is the strongest in the wild-type MmpL3 transporter. In comparison with wild-type MmpL3, the relative binding affinities for T6D in the K528A, H558A, and F561A mutants are 0.023, 2.682, and 3.801 kcal/mol weaker, especially for the mutant transporters H558A and F561A. The data suggest that residues H558 and F561 may be significant for substrate binding and transport.

Interestingly, in the periplasmic region, 4 phenylalanines, F134, F445, F449, and F452, create a hydrophobic patch at the periplasmic subdomain PD1. During the simulation, T6D flipped its hydrophobic tail to tightly interact with this phenylalanine cluster. S9 Fig depicts the initial orientation (at 0 ns) and orientation after 4,000 ns simulation of T6D, suggesting that lipid transport may require a switch in orientation of the bound lipid, where the positively charged residues R523, K528, and H558 and hydrophobic residues F134, F445, F449, and F452 are critical to guide this flipping.

To further elucidate the mechanism of substrate transport, we performed target MD simulations using the NAMD program. First, we observed that the trehalose headgroup of T6D interacts with the charged and polar residues H558, Q554, E553, N524, and S423 of the T6D-binding site between TMs 7 to 10. During the transfer of T6D to the periplasmic T6D-binding site, the region of the MmpL3 channel formed between 2 flexible loops (constituting residues R523-D525 on one side and S423-G426 on the other side of the channel) gradually open up and allow for T6D to reach the cavity between subdomains PD1 and PD2 (Fig 7). The minimum distances between the 2 loops as a function of time is shown in S10 Fig.

## Discussion

Translocation of TMM lipids to the mycobacterial cell wall is an essential step for cell wall biogenesis. Despite the importance of this process for *M. tuberculosis* virulence and pathogenesis, the mechanism on how MmpL3 facilitates the shuttling of lipid substrates is still unclear. In this study, we successfully obtained both cryo-EM and X-ray structures of the *M. smegmatis* MmpL3 transporter at various conformational states. These structural differences allowed us to directly observe multiple transient states that MmpL3 may need to go through during lipid transport (Figs 5 and 6A). Our structural data are in line with MD simulations that indicate

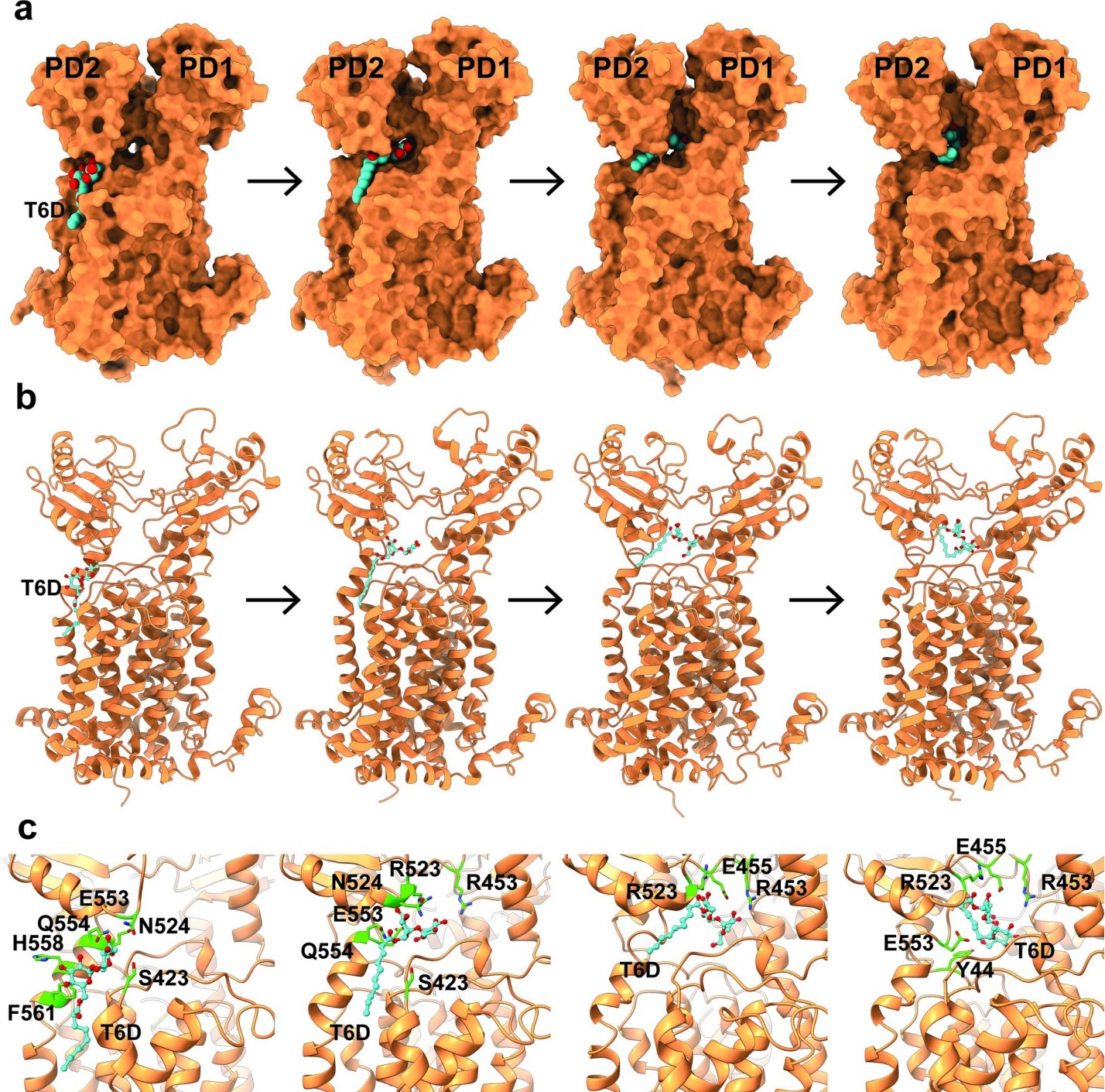

**Fig 7. Target MD simulations of the MmpL3 transporter.** The calculations depict snapshots (0, 1.26, 2.27, and 3.11 ns) of T6D shuttling from the outer leaflet of the inner membrane, between TMs 7–10, to the periplasmic central cavity, between PDs 1–2, of MmpL3. MD, molecular dynamics; MmpL3, mycobacterial membrane protein large 3; T6D, trehalose 6-decanoate.

the major conformational change of the transporter is located at the periplasmic subdomain PD2, where this movement can easily be interpreted as a rigid body rotational motion of PD2 in relation to the PD1 subdomain (Fig 6A and 6B). This motion provides the plasticity of the

periplasmic central cavity between PD1 and PD2, where the volume of the cavity can be adjusted to accommodate lipid binding.

It has been proposed that MmpL3 shuttles TMM from the outer leaflet of the cytoplasmic membrane via the substrate binding site formed by TMs 7 to 10. The TMM lipid is then transferred to the periplasmic central binding cavity through the channel created by MmpL3 [14]. It appears that the flexible loops constituting residues 523 to 525 and residues 423 to 426 form the gate for the opening and closing of the channel. Based on the structural information from both cryo-EM and X-ray crystallography, it is observed that there is a long-distance coupling interaction between the periplasmic and transmembrane domains of MmpL3. It is likely that the conformational changes of TMs 7 to 10 and PD2, as well as the alteration of the distance between TM2 and TM8, are coordinated to facilitate TMM transport from the outer leaflet of the transmembrane to the periplasmic domains of MmpL3.

The X-ray structures of MmpL3 bound with a variety of inhibitors indicate that TMs 4, 5, 6, 10, 11, and 12 of this transporter participate in forming a large binding pocket to accommodate for the binding of these inhibitors [15]. Each of these small molecules were found within the middle of the transmembrane domain, presumably disrupting the proton relay network of the transporter. However, our cryo-EM and X-ray structures of MmpL3, either alone or bound with lipid moieties, depict that the TM helices are closely packed and do not appear to form a cavity or pocket to bind substrates. There is a chance that the inhibitor-binding site at the transmembrane domain could open up only in the presence of these inhibitors. Therefore, it is unlikely that the MmpL3 transporter would utilizes this site as a pathway to shuttle lipid moieties.

Our computational simulations further suggest that subdomain PD1 of MmpL3 makes up a hydrophobic patch, which contains residues F134, F445, F449, and F452 (S9 Fig). Similar hydrophobic patches have been found in other bacterial multidrug efflux pumps, including AcrB [19], AdeB [20,21], and MtrD [22], where the patch residues have been observed to be directly involved in substrate binding. We believe that this MmpL3 hydrophobic patch may be critical for interaction with the hydrocarbon chains of lipids to advance lipid transport. Indeed, F134, F445, and F452 are found to specifically contact the hydrophobic tail of the bound $TMM_2$ molecule in the MmpL3-TMM III structure.

Previously, it was observed that MmpL3 behaves like a flippase [5], which translocates TMM from the inner leaflet to outer leaflet of the cytoplasmic membrane. If this is the case, then the cryo-EM structure of MmpL3-TMM III may capture the conformational states of the transporter after the TMM lipid has been flipped from the inner leaflet to outer leaflet of the membrane (Fig 8). Interestingly, TM8 and TM9 are located at the outermost surface of the MmpL3 transporter, directly interacting with membrane phospholipids. These 2 TM helices create a V-shaped architecture near the cytoplasmic surface. There is a chance that TM8 and TM9 may contribute to facilitate the flipping of TMM molecules within the cytoplasmic membrane by a subtle movement of the V-shaped architecture formed by these 2 TM helices. Indeed, our MmpL3-TMM complex structure indicates that TM8 and TM9 are substantially engaged in anchoring the hydrophobic tail of bound TMM (Fig 4B). Based on the structural and computational information, the bound TMM lipid may need to flip one more time when it is shuttled from the outer leaflet of transmembrane region to the periplasmic domain of this transporter (Figs 4 and 8).

## Methods

### Cloning, expression, and purification of MmpL3 and MmpL3$_{773}$

The procedures of cloning and expression *M. smegmatis* full-length *mmpL3* and *mmpL3$_{773}$* were described previously [14]. The procedures for purifying the MmpL3 and MmpL3$_{773}$ proteins were the same. The collected bacteria expressing MmpL3 or MmpL3$_{773}$ were resuspended

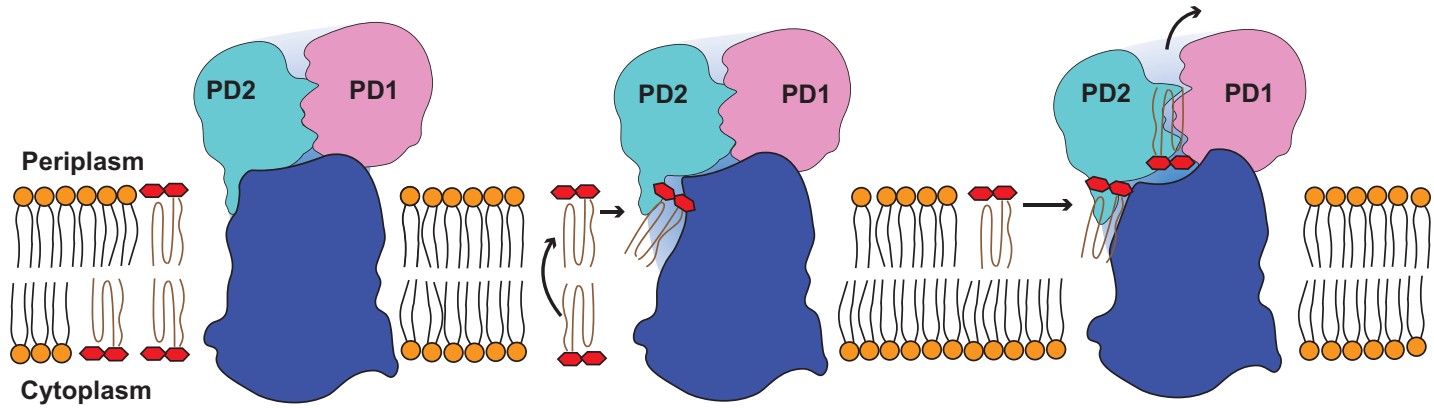

**Fig 8. Proposed mechanism for TMM translocation via MmpL3.** This schematic diagram indicates that the MmpL3 transporter is capable of flipping a TMM molecule from the inner leaflet to outer leaflet of the cytoplasmic membrane. The transporter will pick up this TMM molecule from the outer leaflet of the membrane, allowing the TMM molecule to pass through the channel formed by MmpL3 and arrive at the periplasmic lipid binding site. The orientation of the bound TMM molecule in the periplasmic domain of MmpL3 is antiparallel to that of bound TMM at the outer leaflet of the transmembrane region. MmpL3, mycobacterial membrane protein large 3; TMM, trehalose monomycolate.

in low salt buffer (100 mM sodium phosphate (pH 7.2), 10% glycerol, 1 mM ethylenediaminetetraacetic acid (EDTA), and 1 mM phenylmethanesulfonyl fluoride (PMSF)), and then disrupted with a French pressure cell. The membrane fraction was collected and washed twice with high salt buffer (20 mM sodium phosphate (pH 7.2), 2 M KCl, 10% glycerol, 1 mM EDTA, and 1 mM PMSF), and once with 20 mM HEPES-NaOH buffer (pH 7.5) containing 1 mM PMSF as described previously [23]. The MmpL3 or MmpL3$_{773}$ membrane protein was extracted and delipidated using the established protocol [24]. The protein was then purified with a Ni$^{2+}$-affinity column and dialyzed against 20 mM Na-HEPES (pH 7.5). A final purification step was performed using a Superdex 200 size exclusion column loaded with buffer solution containing 20 mM Na-HEPES (pH 7.5) and 0.05% (w/v) n-dodecyl-β-D-maltoside (DDM).

## Nanodisc embedded sample preparation

To assemble MmpL3 into nanodiscs, a mixture containing 10 μM MmpL3, 30 μM MSP (cNW9), and 900 μM *E. coli* total extract lipid was incubated for 15 minutes at room temperature. After, 0.8 mg/mL prewashed Bio-beads (Bio-Rad, Hercules, California, USA) was added. The resultant mixture was incubated for 1 hour on ice followed by overnight incubation at 4˚C. The protein-nanodisc solution was filtered through 0.22 μm nitrocellulose-filter tubes to remove the Bio-beads. To separate free nanodiscs from MmpL3-ND, the filtered protein-nanodisc solution was purified using a Superose 6 column (GE Healthcare Life Sciences, Marlborough, Massachusetts, USA) equilibrated with 20 mM Tris-HCl (pH 7.5) and 100 mM NaCl. Fractions corresponding to the size of the MmpL3-ND complex were collected for cryo-EM.

For making the MmpL3-TMM samples, 10 μM MmpL3, 30 μM MSP (cNW9) and 900 μM lipid mixture (80% *E. coli* total extract lipid and 20% *M. smegmatis* glycolipid) were mixed together and incubated for 15 minutes at room temperature. The procedures for adding Bio-beads and separating free nanodiscs from the MmpL3-TMM complexes were the same as those for MmpL3-ND.

## GDN embedded sample preparation

To prepare the purified MmpL3 protein in GDN detergent, the MmpL3 membrane protein was extracted and delipidated in buffer containing 20 mM Na-HEPES (pH 7.5), 2% GDN, and

2% octyl glucose neopentyl glycol (OGNG) at 4°C with stirring for 16 hours. Insoluble materials were then removed and separated from the solubilized membrane protein using ultracentrifugation at $100,000 \times g$. The extracted protein was then purified with a $Ni^{2+}$-affinity column, dialyzed against 20 mM Na-HEPES (pH 7.5), and concentrated to 20 mg/ml in a buffer containing 20 mM Na-HEPES (pH 7.5) and 0.01% GDN.

## Cryo-EM data collection

For MmpL3-ND, a 3 μl of 0.2 mg/ml MmpL3-ND sample was applied to glow-discharged holey carbon grids (Quantifoil Cu R1.2/1.3, 300 mesh), blotted for 3 seconds and then plunge-frozen in liquid ethane using a Vitrobot (Thermo Fisher, Waltham, Masachusetts). The grids were transferred into cartridges. The images were recorded at 1 to 3.5 μm defocus on a K2 summit direct electron detector (Gatan, Pleasanton, California) with super-resolution mode at nominal 130 K magnification corresponding to a sampling interval of 1.06 Å/pixel (super-resolution 0.53 Å/pixel). Each micrograph was exposed for 10 seconds with 5 $e^-/A^2$/sec dose rate (total specimen dose, 40 $e^-/A^2$). A total of 40 frames per specimen area were captured using Latitude (Gatan, Pleasanton, California).

A Titan Krios cryo-electron transmission microscope (Thermo Fisher, Waltham, Massachusetts) with a K3 direct electron detector (Gatan, Pleasanton, California) was used to collect cryo-EM images for the MmpL3-TMM and MmpL3-GDN samples. A 3 μl of 0.2 mg/ml MmpL3-TMM sample was applied to glow-discharged holey carbon grids (Quantifoil Cu R1.2/1.3, 300 mesh), blotted for 3 seconds, and then plunge-frozen in liquid ethane using a Vitrobot (Thermo Fisher). The grids were then transferred into cartridges. The images of MmpL3-TMM were recorded at 1 to 2.5 μm defocus with super-resolution mode at a physical pixel size of 0.52 Å/phys. Pixel (super-resolution, 0.26 Å/pixel). Each micrograph was exposed for 2 seconds with 18.2 $e^-$/sec/phys. Pixel dose rate (total specimen dose, 30 $e^-/A^2$). A total of 40 frames per specimen area were captured using SerialEM [25]. For the MmpL3-GDN sample, a 2.5 μl of 5 mg/ml MmpL3-GDN was applied to glow-discharged holey carbon grids (Quantifoil R1.2/1.3 Cu 300 mesh). Data were recorded at a nominal magnification of 81,000 corresponding to a pixel size of 1.08 Å/pixel (super-resolution, 0.54 Å/pixel) with a total exposure time of 2.9 seconds (58 individual frames).

## Cryo-EM data processing

The micrographs of MmpL3 were aligned using a patch-based motion correction for beam-induced motion using cryoSPARC [26]. The contrast transfer function (CTF) parameters of the micrographs were determined using Patch CTF [27]. After manual inspection and sorting to discard poor micrographs, approximately 2,000 particles of MmpL3 were manually picked to generate templates for automatic picking. Initially, 1,016,957 particles were selected after autopicking in cryoSPARC [26]. Several iterative rounds of two-dimensional (2D) classifications were carried to remove false picks and classes with unclear features. The resulting 126,486 particles were used to generate a reference free ab initio three-dimensional (3D) reconstruction with C1 symmetry. After several rounds of heterogeneous refinement, 75,398 particles were chosen for further processing with nonuniform and local CTF refinement [27]. Refinement and reconstruction of the selected particle projections after subtracting densities for the nanodiscs enabled us to obtain a 3D reconstruction of the transporter at 3.65 Å global resolution map based on the gold standard Fourier shell correlation (FSC 0.143). The cryo-EM map was then improved using the RESOLVE density modification program [28] implemented in PHENIX [29] to a final resolution of 3.00 Å.

For MmpL3-TMM, 764,338 particles were initially selected from autopicking in cryoS-PARC [26]. After several rounds of 2D classifications, 163,553 particles were selected for several additional iterative rounds of 3D heterogeneous refinement. The resulting 2 distinct classes were chosen for further processing with nonuniform and local focused refinement. To enhance structural features from the membrane protein, densities corresponding to nanodiscs were subtracted from individual particles. The modified particles of MmpL3-TMM I and MmpL3-TMM II were further refined with local focused refinement, resulting in resolutions of 4.27 Å and 4.33 Å, respectively. The cryo-EM maps were then improved using the RESOLVE density modification program [28] implemented in PHENIX [29] to final resolutions of 3.30 Å for both MmpL3-TMM I and MmpL3-TMM II.

The MmpL3-GDN and MmpL3-TMM III cryo-EM data processing were completed using the same approach as described above, resulting in resolution of 2.94 Å and 2.66 Å, respectively. Their cryo-EM maps were then improved using the RESOLVE density modification program [28] implemented in PHENIX [29] to final resolutions of 2.40 Å and 2.20 Å, respectively.

## Model building and refinement

Model building of MmpL3-ND was based on the 3.65 Å cryo-EM map. The 2.59 Å structural model of MmpL3$_{773}$-PE (pdb id: 6OR2) [14] was used to fit into the density map using Chimera [30]. The subsequent model rebuilding was performed using Coot [31]. Structure refinements were performed using the phenix.real_space_refine program [32] from the PHENIX suite [29]. The final atomic model was evaluated using MolProbity [33]. The statistics associated with data collection, 3D reconstruction, and model refinement are included in S1 Table.

The MmpL3-GDN, MmpL3-TMM I, MmpL3-TMM II, and MmpL3-TMM III structural models were built based on the MmpL3-ND structure. Structural refinements were done using the same approach as described above (S1 Table).

## Crystallization of the MmpL3$_{773}$-T6D complex

Before crystallization, the purified MmpL3$_{773}$ protein (50 μM) was incubated with 250 μM T6D at 25˚C for 2 hours. The MmpL3$_{773}$-T6D crystals were then grown at 25˚C using vapor diffusion. A 2-μl protein solution containing 50 μM MmpL3$_{773}$ and 250 μM T6D in buffer containing 20 mM Na-HEPES (pH 7.5) and 0.05% (w/v) DDM was mixed with 2 μl of reservoir solution containing 25% PEG 400, 0.1 M sodium acetate (pH 5.4), and 0.05 M magnesium acetate. The resultant mixture was equilibrated against 500 μl of the reservoir solution. Cryoprotection was achieved by raising the PEG 400 concentration to 30%.

## X-ray crystallography data collection, structural determination, and refinement

All diffraction data were collected at 100 K at beamline 24ID-C located at the Advanced Photon Source, using a Pilatus 6M detector (Dectris, Switzerland). Diffraction data were processed using DENZO and scaled using SCALEPACK [34]. Crystals of MmpL3$_{773}$-T6D took the space group C2 (S2 Table).

The crystal structure of MmpL3$_{773}$-T6D was determined by molecular replacement (MR), utilizing the crystal structure of MmpL3$_{773}$-PE (pdb id: 6OR2) [14] as a search model. The initial model of MmpL3$_{773}$ was then modified manually using the program Coot [31]. The model was refined using PHENIX [29], leaving 5% of reflections in the Free-R set. Iterations of refinement were performed using PHENIX [29]. Model building was done using Coot [31], which led to the current model (S2 Table).

## Molecular dynamics (MD) simulations

The protonation states of the titratable residues of the MmpL3$_{773}$ transporter was determined using the H++ server (http://biophysics.cs.vt.edu/). Using the MmpL3$_{773}$-T6D I crystal structure as the template, we removed both bound T6D ligands (MmpL3$_{773}$ (apo)), removed the bound T6D located at TMs 7 to 10 (MmpL3$_{773}$ (T6D-single)) and retained both bound T6Ds (MmpL3$_{773}$ (T6D-double)). These 3 structures were separately immersed in an explicit lipid bilayer consisting of POPC and POPE with a molecular ratio of 1:1, and a water box with dimensions of 110.9 Å × 111.3 Å × 150.4 Å using the CHARMM-GUI Membrane Builder webserver (http://www.charmm-gui.org/?doc=input/membrane). We then added 150 mM NaCl and extra neutralizing counter ions for the simulations. The total number of atoms were 141,497, 141,571, and 141,645 for the MmpL3 (apo), MmpL3 (T6D-single), and MmpL3 (T6D-double) systems, respectively. The Antechamber module of AmberTools was employed to generate parameters for T6D by using the general AMBER force field (GAFF) [35,36]. The partial charges of T6D were calculated using ab initio quantum chemistry at the HF/6-31G$^*$ level (GAUSSIAN 16 program) (Gaussian, Wallingford). The RESP charge-fitting scheme was used to calculate partial charges on the atoms [37]. The tleap program was used to generate parameter and coordinate files using the ff14SB and Lipid17 force field for both the protein and lipids. The PMEMD.CUDA program implemented in AMBER18 (AMBER 2018, UCSF) was used to conduct MD simulations. The simulations were performed with periodic boundary conditions to produce isothermal–isobaric ensembles. Long-range electrostatics were calculated using the particle mesh Ewald (PME) method [38] with a 10-Å cutoff. Prior to the calculations, energy minimization of these systems was carried out. Subsequently, the systems were heated from 0 K to 303 K using Langevin dynamics with the collision frequency of 1 ps$^{-1}$. During heating, the MmpL3$_{773}$ transporter was position restrained using an initial constant force of 500 kcal/mol/Å$^2$ and weakened to 10 kcal/mol/Å$^2$ to allow for the movement of lipid and water molecules. Then, the systems went through 5 ns equilibrium MD simulations. Finally, a total of 4 μs production MD simulations were conducted. During simulations, the coordinates were saved every 500 ps for analysis. All systems were well equilibrated after 1 μs simulations according to RMSDs of the transporter Cα atoms. Approximately 1 to 4 μs trajectories of each system were used for root mean square fluctuation (RMSF) and PCA [39,40]. GROMCAS analysis tools were used for the MD simulation trajectory analysis [41]. The MM-GBSA module [18] implemented in AMBER18 (AMBER 2018, UCSF) was used to perform binding free energy calculations in the computational alanine scanning experiment. The binding free energy between MmpL3$_{773}$ and T6D consists of the following energy terms: nonbonded electrostatic interaction, van der Waals energy in gas phase, polar, and nonpolar solvation free energy. The polar component is calculated using Generalized Born (GB) implicit solvation model. The nonpolar component is calculated using solvent accessible surface area model. A total of 100 snapshots were extracted from the 420 to 520 ns along the MD simulation trajectory at an interval of 1 ns for the alanine scanning binding free energy calculations.

## Target MD simulations

Target MD (TMD) was performed using the NAMD program [42] with the same AMBER force field parameters as described above. In the simulations, we selected the heavy atoms of the T6D lipid headgroup to be guided toward the target position at the periplasmic lipid-binding site by the application of steering forces. The root mean square (RMS) distance between the current coordinates and the target structure was calculated at each timestep. The force on each selected atom was given by a gradient of potential as a function of the RMS values. The system was gone through energy minimization, heating, and 5 ns equilibrium MD simulations.

Then, TMD simulation was performed for 5 ns based on the MD equilibrated coordinates. A value of 500 kcal/mol/Å$^2$ was used as an elastic constant for TMD forces during the simulations.

## Accession codes

Atomic coordinates and structure factors have been deposited with accession codes 7K8A (PDB) and EMD-22724 (EMBD) for MmpL3-ND; 7K8B (PDB) and EMD-22725 (EMDB) for MmpL3-GDN; 7K8C (PDB) and EMD-22726 (EMDB) for MmpL3-TMM I; 7K8D (PDB) and EMD-22728 (EMDB) for MmpL3-TMM II; 7N6B (PDB) and EMD-24206 (EMDB) for MmpL3-TMM III; and 7K7M (PDB) for MmpL3$_{773}$-T6D I and MmpL3$_{773}$-T6D II.

This work is based upon research conducted at the Northeastern Collaborative Access Team beamlines, which are funded by the National Institute of General Medical Sciences from the National Institutes of Health (P30 GM124165). This research used resources of the Advanced Photon Source, a U.S. Department of Energy (DOE) Office of Science User Facility operated for the DOE Office of Science by Argonne National Laboratory under Contract No. DE-AC02-06CH11357.

## Supporting information

**S1 Table. Cryo-EM data collection, processing, and refinement statistics.**
(TIF)

**S2 Table. X-ray data collection and structural refinement statistics.**
(TIF)

**S1 Fig. Cryo-EM analysis of MmpL3-ND.** (a) Data processing flow chart with particle distributions. The black box indicates the particle class used for further refinement. (b) Representative 2D classes. (c) FSC curves. (d) Viewing direction distribution calculated in cryoSPARC for particle projections. This heat map shows number of particles for each viewing angle. (e) Sharpened cryo-EM map of the MmpL3 transporter viewed in the membrane plane. (f) Local EM density map of MmpL3. cryo-EM, cryo-electron microscopy; CTF, contrast transfer function; FSC, Fourier shell correlation; MmpL3, mycobacterial membrane protein large 3; MmpL3-ND, MmpL3-nanodisc.
(TIF)

**S2 Fig. Superimposition of the cryo-EM structure of MmpL3-ND to that of the X-ray structure of MmpL3$_{773}$-PE.** The superimposition gives rise to an RMSD of 1.5 Å (MmpL3-ND, orange; MmpL3$_{773}$-PE, cyan). cryo-EM, cryo-electron microscopy; MmpL3-ND, MmpL3-nanodisc; RMSD, root mean square deviation.
(TIF)

**S3 Fig. Cryo-EM analysis of MmpL3-GDN.** (a) Data processing flow chart with particle distributions. The black box indicates the particle class used for further refinement. (b) Representative 2D classes. (c) FSC curves. (d) Viewing direction distribution calculated in cryoSPARC for particle projections. This heat map shows number of particles for each viewing angle. (e) Sharpened cryo-EM map of MmpL3-GDN viewed in the membrane plane. (f) Local EM density map of MmpL3-GDN. cryo-EM, cryo-electron microscopy; CTF, contrast transfer function; FSC, Fourier shell correlation; MmpL3-GDN, MmpL3-glycol-diosgenin.
(TIF)

**S4 Fig. Cryo-EM analysis of MmpL3-TMM I and MmpL3-TMM II.** (a) Data processing flow chart with particle distributions. The black and red boxes indicate the particle classes used for

further refinement of the structures of MmpL3-TMM I and MmpL3-TMM II, respectively. (b) Representative 2D classes. (c) Viewing direction distribution calculated in cryoSPARC for MmpL3-TMM I particle projections. The heat map shows number of MmpL3-TMM I particles for each viewing angle. (d) Viewing direction distribution calculated in cryoSPARC for MmpL3-TMM II particle projections. The heat map shows number of MmpL3-TMM II particles for each viewing angle. (e) FSC curves for the structure of MmpL3-TMM I. (f) Sharpened cryo-EM map of MmpL3-TMM I viewed in the membrane plane and its local EM density map. (g) FSC curves for the structure of MmpL3-TMM II. (h) Sharpened cryo-EM map of MmpL3-TMM II viewed in the membrane plane and its local EM density map. cryo-EM, cryo-electron microscopy; CTF, contrast transfer function; FSC, Fourier shell correlation; MmpL3-TMM, MmpL3-trehalose monomycolate.
(TIF)

**S5 Fig. Electron density maps of MmpL3$_{773}$-T6D I and MmpL3$_{773}$-T6D II at a resolution of 3.34 Å.** The electron density maps are contoured at 1.2 σ. The Cα traces of the 2 MmpL3 molecules in the asymmetric unit are in red (MmpL3$_{773}$-T6D I) and green (MmpL3$_{773}$-T6D II). These MmpL3 structures were determined by MR, utilizing the MmpL3$_{773}$-PE structure (pdb id: 6OR2) as a search model. MmpL3, mycobacterial membrane protein large 3; MR, molecular replacement.
(TIF)

**S6 Fig. Cryo-EM analysis of MmpL3-TMM III.** (a) Data processing flow chart with particle distributions. The black box indicates the particle class used for further refinement. (b) Representative 2D classes. (c) FSC curves. (d) Viewing direction distribution calculated in cryoSPARC for particle projections. This heat map shows number of particles for each viewing angle. (e) Sharpened cryo-EM map of MmpL3-TMM III viewed in the membrane plane. (f) Local EM density map of MmpL3-TMM III. cryo-EM, cryo-electron microscopy; CTF, contrast transfer function; FSC, Fourier shell correlation; MmpL3-TMM, MmpL3-trehalose monomycolate.
(TIF)

**S7 Fig. MD simulations suggest 4 important residues for directing the transport of T6D from the transmembrane to periplasmic lipid-binding sites.** (a) Minimum distances between T6D and these 4 residues (T6D-R523, black; T6D-K528, red; T6D-H558, green; and T6D-F561, blue). (b and c) The locations of residues R523, K528, H558, and F561. Residues R523, K528, H558, and F561 are shown as yellow sticks. The bound T6D molecule is in green sticks. The secondary structural elements of MmpL3 are colored pink. MD, molecular dynamics; MmpL3, mycobacterial membrane protein large 3; T6D, trehalose 6-decanoate.
(TIF)

**S8 Fig. Protein sequence alignment for 6 different mycobacterial MmpL3 transporters.** This alignment suggests that the gate residues S423, L424, and D525, as well as the residues K528, H558, and F561 that interact with T6D are conserved among these transporters. MmpL3, mycobacterial membrane protein large 3; T6D, trehalose 6-decanoate.
(TIF)

**S9 Fig. The hydrophobic Phe cluster of MmpL3.** (a) MD simulations indicate the minimum distance between T6D and the Phe cluster as a function of time. (b) The locations of T6D (especially the hydrocarbon tail) and the Phe cluster (F134, F449, F452, and F455) at 0 ns. The T6D molecule is colored cyan. Residues F134, F449, F452, and F455 are colored green. The secondary structural elements of MmpL3 are colored orange. (c) The locations of T6D

(especially the hydrocarbon tail) and the Phe cluster (F134, F449, F452, and F455) at 4,000 ns. The T6D molecule is colored cyan. Residues F134, F449, F452, and F455 are colored green. The secondary structural elements of MmpL3 are colored orange. MmpL3, mycobacterial membrane protein large 3; T6D, trehalose 6-decanoate.
(TIF)

**S10 Fig. TMD simulations of the gate formed by the flexible loops S423-G426 and R523-D525.** This figure indicates the minimum distance of the gate as function of time during TMD simulations. MD, molecular dynamics; TMD, target MD.
(TIF)

## Author Contributions

**Conceptualization:** Chih-Chia Su, Philip A. Klenotic, Meng Cui, Edward W. Yu.

**Data curation:** Chih-Chia Su, Philip A. Klenotic, Meng Cui, Edward W. Yu.

**Formal analysis:** Chih-Chia Su, Philip A. Klenotic, Meng Cui, Edward W. Yu.

**Funding acquisition:** Edward W. Yu.

**Investigation:** Chih-Chia Su, Philip A. Klenotic, Meng Cui, Meinan Lyu, Christopher E. Morgan, Edward W. Yu.

**Methodology:** Chih-Chia Su, Philip A. Klenotic, Meng Cui, Meinan Lyu, Christopher E. Morgan, Edward W. Yu.

**Project administration:** Edward W. Yu.

**Supervision:** Chih-Chia Su, Edward W. Yu.

**Validation:** Chih-Chia Su, Philip A. Klenotic, Meng Cui, Edward W. Yu.

**Visualization:** Chih-Chia Su, Philip A. Klenotic, Meng Cui.

**Writing – original draft:** Edward W. Yu.

**Writing – review & editing:** Philip A. Klenotic, Meng Cui, Edward W. Yu.

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
