## [Editor Report · Decision Letter 0]

12 Feb 2021

Dear Dr. Yu, 

Thank you for submitting your manuscript entitled "Structures of MmpL3 illuminate a mechanism of lipid transport" for consideration as a Research Article by PLOS Biology.

Your manuscript has now been evaluated by the PLOS Biology editorial staff, as well as by an academic editor with relevant expertise, and I am writing to let you know that we would like to send your submission out for external peer review.

Please re-submit your manuscript within two working days, i.e. by Feb 14 2021 11:59PM.

Kind regards,

Paula

---

Associate Editor

PLOS Biology

---

## [Decision Letter · Decision Letter 1]

9 Apr 2021

Dear Dr. Yu,

Thank you very much for submitting your manuscript "Structures of MmpL3 illuminate a mechanism of lipid transport" for consideration as a Research Article at PLOS Biology. Your manuscript has been evaluated by the PLOS Biology editors, an Academic Editor with relevant expertise, and by several independent reviewers.

In light of the reviews (below), we are pleased to offer you the opportunity to address the comments from the reviewers in a revised version that we anticipate should not take you very long. We will then assess your revised manuscript and your response to the reviewers' comments and we may consult the reviewers again.

In particular, reviewer #1 thinks that you should discuss more previous MmpL3 structures, and asks what is new/significant about the sites presented here, also asks about the structural differences observed among cryoEM nanodisc MmpL3 structures, asks how are cavity volumes calculated and how are structures superposed, and suggests to do experiments utilizing a wider range of TMM, in multiple concentration steps, with effect upon observed structures and experiments with T6D instead of TMM. This reviewer also thinks that your assumptions and conclusions are highly speculative, says that the discussion should be shortened and clarified, and it would be useful to have specific functional predictions made for the effects of specific mutations that are suggested by the structures, analysis and simulations, and multiple sequence alignment (MSA) of mycobacterial MmpL3 orthologs. Reviewer #2 says that a scheme of the process of TMM translocation would be very helpful for the reader, finds the description of the protein expression and purification to be inaccurate, thinks that you should mention that RMSD is an all atom RMSD or a backbone, asks why you think the conformational change seem to be always away from the membrane, wants you to discuss your model where MmpL3 is not a flippase and whether there is another flippase associated, and says it would be nice to include an Euler angle distribution of all particles used in the final map reconstruction in the supplementary figures.

We expect to receive your revised manuscript within 2 months.

**IMPORTANT - SUBMITTING YOUR REVISION**

*Resubmission Checklist*

*Published Peer Review*

*PLOS Data Policy*

*Blot and Gel Data Policy*

Sincerely,

Paula

---

Associate Editor,

pjaureguionieva@plos.org,

PLOS Biology

REVIEWS:

Reviewer #1: Structural Biology of transports

Reviewer #2: Structural biology of membrane proteins

Reviewer #1: The mycobacterial membrane transport protein MmpL3 is of both fundamental and practical significance, as it is essential for transport of trehalose monomycolate (TMM), a precursor for several components of the mycobacterial outer membrane. It is a target of several tuberculosis (TB) drugs, and a validated target for future drug discovery/development. Although no M. tuberculosis (Mtb) MmpL3 structures have been determined to date, Mtb MmpL3 can be complemented by the MmpL3 ortholog from M. smegmatis (Msm). The first structures of Msm MmpL3 were determined in 2019, first by Zhang et al. (Cell 176:636), and then shortly thereafter, also in 2019, by the Yu group (PNAS 116:11241), who are the authors of this PLoS Biology submission. This manuscript presents and describes six structures, determined by x-ray crystallography or cryoelectron microscopy (these latter being the first cryoEM structures of MmpL3), accompanied by detailed analysis of the structures along with several types of atomistic molecular dynamics (MD) simulations. The structures presented differ in a variety of ways, and the goal of the subsequent analysis, and especially of the MD simulations, is to infer/suggest/hypothesize transport mechanisms of MmpL3 based upon these structural "inputs."

The PI has a distinguished record of excellence in membrane protein structural biology, with a focus upon x-ray crystallography of bacterial AcrB transporters. The MmpL3 structures presented are a welcome addition and will likely be of significant future utility in the design of functional experiments (including MmpL3 mutagenesis) to obtain detailed mechanistic insights into the transport cycle of this important mycobacterial transporter. However, to this Reviewer, the mechanism suggested here, of coupled movement of the MmpL3 periplasmic domain with its transmembrane (TM) helices, is rather highly speculative and not compellingly demonstrated by the combination of structures, structural analysis and MD simulations presented. Some specific questions/concerns follow.

1. Previous MmpL3 structures are insufficiently discussed, and bear upon the analysis presented. The structures of Zhang et al. (Cell 176:636) include MmpL3 with bound Tb drugs/drug candidates, where these appear to be located in the lumen of the transporters. Do the authors of this manuscript consider this site to also be in the transport/permeation pathway of the TMM? Even more critically, the authors' previous MmpL3 structural publication (PNAS 116:11241) indicates that the presence of bound detergent (DDM) and bound lipid (PE). To this Reviewer's eye, the DDM site looks like the "T6D1" site & the PE site looks like the "T6D2" site in the structures presented in this manuscript, respectively. Therefore, what is (most) new/significant about the sites presented here?

2. Are the structural differences observed among cryoEM nanodisc MmpL3 structures due to membrane variation caused by addition of TMM to the "membrane mimetic", rather than arising from functional conformational substates of the transporter? MmpL3 nanodiscs (NDs) were prepared in the absence and presence of Msm glycolipids (btw, details on preparation/sourcing of these glycolipids appears to be absent). No (ordered) TMM is seen in the cryoEM structures of the +TMM ND samples, which seems suggestive of TMM mixing with the E. coli lipids to form the bilayer region encircled by the molecular scaffold protein (MSP), rather than binding to MmpL3 (T6D was seen in the detergent-solubilized x-ray structure). Were there any characteristics of the NDs, such as diameter or thickness, which were different +/- TMM? Future experiments utilizing a wider range of TMM, in multiple concentration steps, with effect upon observed structures could help sort this out. Also, experiments with T6D instead of TMM could be very informative in ND cryoEM structures, in terms of actually showing up as bound to MmpL3. The variation of MmpL3 in ND vs. GDN is suggestive of different membrane mimetics affecting structures.

3. A primary result obtained from the structures is that periplasmic subdomain PD2 is in variable positions relative to periplasmic subdomain PD1. This is a solid structural/empirical observation. However, what follows from this, to this Reviewer, is very highly speculative. While the speculative nature of this is mentioned reasonably well at some places in the manuscript, some of the assumptions (and conclusions) are presented with a much higher level of certainty. 

- "This rigid-body rotation may allow for the expansion of the central periplasmic cavity ..."

- "These two major conformational changes may couple with each other ..."

- "... these six MmpL3 structures can be interpreted as snapshots of various transient states ..."

(this last statement seems especially unsubstantiated.)

- "It is likely that S423 and N254 form a gate of the channel that is able to open and close ..."

The relation of the MD results, at shorter length scales, to the larger-scale structural variations observed and the couplings mentioned, are not connected in a way that is clear to this Reviewer. They seem more like rather independent results. This discussion could be shortened and clarified. It would be useful to have specific functional predictions made for the effects of specific mutations that are suggested by the structures, analysis and simulations. Also, multiple sequence alignment (MSA) of mycobacterial MmpL3 orthologs could add to this.

4. How are cavity volumes calculated? How are structures superposed, for rmsd?

Reviewer #2: The MmpL3 is an essential protein in mycobacterium. It is required for shuttling TMM from the cytoplasmic membrane out to the outer membrane of mycobacterium. The authors Su et al here present 6 different structures of MmpL3; in detergent, in nanodisc, with and without ligands. Based on these structures the authors are able to speculate as to how substrate is bound and transferred from the lipid bilayer to the periplasmic binding pocket of the protein. The work is complemented with MD simulations further illustrating the movements within the protein upon substrate binding. Overall, I think this is a comprehensive and very informative study, that I think would be a great addition to PLOS Biology. 

I do have a couple of suggestions that I think would make the manuscript easier to follow, and a couple of questions for the authors to address. 

* A schematic of the process of TMM translocation would be very helpful for the reader. For example a cartoon drawing showing the inner and outer membrane, the position of MmpL3 and path of TMM. 

* In the text when the authors describe the protein expression and purification they say "The MmpL3 membrane protein was overproduced in E. coli BL21(DE3)�acrB cells and purified using a Ni2+-affinity column. We delipidated and reconstituted the purified MmpL3 transporter into lipidic nanodiscs." After looking through their methods I believe they delipidate and solubilize before Ni affinity purification and the sentence should therefore be "The MmpL3 membrane protein was expressed in E. coli BL21(DE3)�acrB cells, solubilized using detergent and purified using a Ni2+-affinity column. The purified protein was reconstituted into lipidic nanodiscs." 

* When the authors mention RMSD they should include if this is an all atom RMSD or a backbone. 

* The conformational change between MmpL3 in ND and detergent is relatively small, but I was surprised to see the conformational changes that were observed all seem to be away from the membrane, why do they authors think this conformational change is observed?

* In the discussion the authors propose a model where TMM comes into the MmpL3 protein from the periplasmic side of the cytoplasmic membrane and is transported into the periplasmic pocket. This model would suggest that MmpL3 is in fact not a flippase. I would like a discussion of this in the paper, is there another flippase associated? 

* It seems to me you have some stretching of the density in the ND structure. It would be nice to include an Euler angle distribution of all particles used in the final map reconstruction in the supplementary figures. 

Minor comment: 

Typo in supplementary figure 1, 3 and 4, I should be refinement not refinment. 

It would be helpful with more references to figures and more precise references such as instead of Fig 3 say Fig 3a and so on.

---

## [Decision Letter · Decision Letter 2]

7 Jul 2021

Dear Dr. Yu,

Thank you for submitting your revised Research Article entitled "Structures of MmpL3 illuminate a mechanism of lipid transport" for publication in PLOS Biology. I have now obtained advice from the original reviewers and have discussed their comments with the Academic Editor. 

Based on the reviews, we will probably accept this manuscript for publication, provided you satisfactorily address the following data and other policy-related requests.

DATA POLICY:

Regardless of the method selected, please ensure that you provide the individual numerical values that underlie the summary data displayed in the following figure panels as they are essential for readers to assess your analysis and to reproduce it: Figures S1C, S3C, S4EG, S6C, S10.

**Please also ensure that figure legends in your manuscript include information on where the underlying data can be found, and ensure your supplemental data file/s has a legend.**

We suggest you to modify the title to make it more informative. We suggest: "Structures of the mycobacterial membrane protein MmpL3 reveal its mechanism of lipid transport", but please feel fre to modify it further.

We expect to receive your revised manuscript within two weeks.

*Published Peer Review History*

*Early Version*

Sincerely,

Paula

---

Associate Editor,

pjaureguionieva@plos.org,

PLOS Biology

Reviewer remarks:

Reviewer #1: I consider the revised manuscript "Structures of MmpL3 illuminate a mechanism of lipid transport" by Su et al. to be acceptable for publication. The authors have conscientiously considered the criticisms raised by both Reviewers, and have adequately addressed them. Especially gratifying is that their acting upon a Reviewer comment to increase TMM concentration in nanodisc preparations used for cryoEM samples led to an additional/new MmpL3 structure at quite high resolution --- which contributes significantly to the mechanistic understanding of function which they are seeking.

Thank you for the opportunity to review a manuscript of interesting and impactful science, which was improved substantively by the review process.

Reviewer #2: The authors have sufficiently addressed my previous comments.

---

## [Editor Report · Decision Letter 3]

21 Jul 2021

Dear Dr. Yu,

On behalf of my colleagues and the Academic Editor, Matthew Waldor, I am pleased to say that we can in principle offer to publish your Research Article "Structures of the mycobacterial membrane protein MmpL3 reveal its mechanism of lipid transport" in PLOS Biology, provided you address any remaining formatting and reporting issues. These will be detailed in an email that will follow this letter and that you will usually receive within 2-3 business days, during which time no action is required from you. Please note that we will not be able to formally accept your manuscript and schedule it for publication until you have made the required changes.

PRESS

Sincerely, 

Paula

---

Paula Jauregui, PhD 

Associate Editor 

PLOS Biology
